# One and the same? How similar are basic human values and economic preferences

**Mario Scharfbillig** [1]*, **Jan Cieciuch**[2,3], **Eldad Davidov**[3,4]

**1** Joint Research Centre, European Commission, Brussels, Belgium, **2** Cardinal Stefan Wyszynski University Warsaw, Warsaw, Poland, **3** Departments of Sociology, University of Zurich, Zurich, Switzerland, **4** Institute of Sociology and Social Psychology, University of Cologne, Cologne, Germany

* Marioscharfbillig@gmail.com

**Data Availability Statement:** All data are fully available without restriction. https://osf.io/v3kqt/.

**Funding:** The authors received no specific funding for this work.

## Abstract

Both the basic human values approach and the economic preference approach have been developed and applied to represent fundamental drivers of human behavior in various domains by measuring people's underlying preferences and motivations. Both of them have been used, however, in isolation from each other, the former primarily in social psychology studies and the latter mainly in economic studies. But how similar are they? Finding that they differ may suggest that combining them to explain human behavior might be beneficial. To the best of our knowledge, only a few studies have attempted to explore and empirically examine the theoretical and empirical link between variables in both approaches. The current study tries to fill this gap by examining relations between basic human values and major economic preferences. We examine the associations between the values of self-transcendence, self-enhancement, openness to change and conservation, and the economic preferences of risk aversion (or seeking), altruism, trust, and positive and negative reciprocity. We propose mechanisms as to how they may be associated with one another. For example, we expect an association between conservation and risk aversion as both are motivated by attributing importance to stability and the status quo, or between self-transcendence and altruism, as both are motivated by concern for others. For the empirical analysis we employed convenience samples collected in Poland and Germany. Results in both samples support our expectations: several values and economic preferences are linked in theoretically predictable ways, but only to a weak or moderate extent. We conclude that they are not mutually exclusive but may rather be complementary, and therefore likely both relevant for investigations into explaining behavior.

## Introduction

In the last decades, the two disciplines of economics and social psychology have advanced our understanding of individual choices and behavior using quite different paths. In economics, traditional preference concepts along with newer ones from behavioral economics such as risk preferences, altruism, trust, and positive or negative reciprocity have been used to explain and predict individual behavior under different circumstances, subject to prices, budget

**Competing interests:** The authors have declared that no competing interests exist.

constraints, and strategic considerations [1]. In the following, we use the abbreviated term "economic preferences" to aggregate these types of preferences often used in economics, while acknowledging that they are in fact human preferences that are also studied in other scientific fields. In social psychology, numerous theories have been developed and applied to explain behavior, with values playing an important role among these theories [2], and with the theory of basic human values developed by Schwartz [3] and its extension [4] being one of the most prominent ones. This values theory has been applied successfully in many studies to explain individual attitudes and behavior (for a review, see [5], inter alia). Indeed, this theory is one of the most cited theories in social psychology. It has been suggested that values, which represent abstract principles and goals in life, are likely to influence the formation of attitudes or behavioral patterns that coincide with the values' basic motivations.

Both approaches have been used, however, in isolation from each other in order to explain behavior. But how similar are they? Finding that they differ may suggest that combining them to explain behavior might be beneficial to both disciplines. To the best of our knowledge, only a few studies have tried to explore and empirically examine the theoretical and empirical link between variables in both approaches. Indeed, as Box-Steffensmeier et al. [6] state, "Genuine progress in understanding human behaviour can only be achieved through a multidisciplinary community effort." In the past years, rapid progress has been made in economics in detecting robust relations between changeable factors such as norms, incentives, or information and behavior, and their role in explaining behavior [7]. However, understanding behavior also requires knowledge of what people actually want, desire and value most [8], which are basic elements in psychological value theories.

The basic assumptions underlying the literature on economic preferences and basic human values are quite different. Values are considered as broad trans-situational motivational goals that vary across individuals, are rather stable (although they may change slowly and under certain conditions over time) and have to be "applied" to a given situation [9]. Economic preferences are understood as being more concrete and permanent or only changeable over lifespans [10, 11, 12]. Thus, whereas the economic preferences perspective would assume a fixed relation between outcomes and preferences, the human values approach would assume that the relation between goals and behavior may vary in different contexts. Including both approaches in an attempt to explain behavior might thus help demonstrate whether values, preferences—or both—are at the core of specific behaviors or attitudes [13]. In this paper, we try to take a deeper look at the relationship between basic human values and economic preferences in a first attempt to examine their degree of complementarity. The findings may, in the future, be beneficial for a more conclusive modeling of a behavioral explanation which considers both approaches.

Underlying values or economic preferences are, by nature, difficult to pin down and measure [14, 15], but the approach used to arrive at them is different in the two disciplines. Values are derived in a top-down approach, by describing a person with abstract goals in life and asking individuals how similar they are to this person. By way of contrast, the economic approach operates in the opposite way, starting from the behavior and trying to infer the preference structure using a revealed preference approach through the choices people make. Therefore, bringing these two strands together not only adds to the conceptual thinking of each, but it also shows the relative coherence of the different methodologies, where they converge and where they do not.

In the following sections, we will present the two approaches and derive hypotheses about possible links between economic preferences and basic human values. Next, we will present the data followed by a test of our hypotheses. We will examine how the two relate to each other and discuss the findings.

## Theoretical considerations

### Basic human values

Basic human values are considered a major element of individual identity and in human life [16]. In Schwartz's model they are defined as "trans-situational goals, varying in importance, that serve as guiding principles in the life of a person or other social entity." In 1992, and relying on earlier approaches [17, 18], Schwartz developed a theory specifying the meaning, catalog and measurement of values. The first version of the theory from 1992 proposed 10 values and the revised version of the theory included 19 more specific values [4, 19]. In both versions of the theory there are four higher-order values: self-transcendence, self-enhancement, conservation, and openness to change. Self-enhancement, which reflects achievement or power values, stands in opposition to self-transcendence, which reflects the values of universalism or benevolence. Openness to change, which includes the values of stimulation or self-direction, stands in opposition to conservation values, which include tradition, security, and conformity. The four higher-order basic human values are underlined by different motivations underlying human behavior. People who attribute high importance to self-transcendence values are concerned with the welfare and interests of others. By way of contrast, people attributing high importance to self-enhancement values find it important to pursue their own interests and relative success and dominance over others. Individuals scoring high on openness to change values find autonomy, independence of thought and action or excitement in life important. Conservation stands in opposition to openness to change. People for whom conservation is important emphasize safety, acceptance of customs and restraint of actions or impulses that may violate expectations of others' order, self-restriction, preservation of the past, and resistance to change [7]. The source of these values is, according to theory, embedded in three universal needs of individuals: needs as biological organisms, need for coordinated social interaction, and survival and welfare needs of groups [4]. The model has been tested and validated in hundreds of samples and studies across different countries and continents [20–25].

The theory also suggests that the values be arranged in a (semi-)circle which reflects relationships between the values. This structure is observed not only among younger and older adults, but even among adolescents or preschool children [26]. When values in the circle are opposite one another, pursuing these values at the same time may create a conflict. When their motivations are similar, values are located closer to each other on the circle and pursuing them simultaneously may be compatible. These values have been shown to be associated with and influence a wide range of attitudes (e.g. toward immigration, political orientation, or identifying as a European, to name but a few (see, inter alia, [27–33]) and behaviors [13, 29, 33–35].

In the current study we focus on the four higher-order values. Schwartz et al. [4] suggested that we may use broader or more specific values, depending on our research purpose. This study focused on the higher-order values, because as explained above they entail the basic motivations of the theory. Nevertheless, we also present the empirical analysis on the lower level of values in the Supporting Information.

### Economic preferences

In the current study we are examining potential relations between the four aforementioned higher-order basic human values and the following key and commonly used economic preferences captured in the Global Preference Survey [8]: risk taking, trust, altruism, and negative and positive reciprocity. The survey also contains the preference for patience, but as we did not expect, hypothesize or pre-register any theory-derived relations between patience and any value. Thus, we did not include patience in our analysis.

Risk seeking (or risk aversion) is usually defined based on the expected utility people get from different outcomes that involve uncertainty. For example, a risk-neutral individual would be indifferent about choosing between receiving $100 for certain, or an equal probability of receiving $50 *or* $150. A risk-averse individual would tend to select the safe option ($100), and a risk-seeking individual might prefer the thrill of choosing the risky option ($50 or $150 with equal probabilities) [36]. The concepts of risk aversion and risk seeking have played a prominent role among economists, because they were able to explain many phenomena, like the risk premium of stocks, variations in consumption and career paths, and potentially even smoking [37]. The modeling of risk in economics has gone through a transition based on findings in behavioral economics, away from orientation towards wealth alone, and towards loss aversion relative to specific reference points [38, 39]. In essence, willingness to take a risk entails a person's evaluation of a certain outcome relative to an uncertain outcome. Similar choices in more or less similar forms are present in everyday life, for example in career choices, decisions to immigrate, parking, etc. [40, 41], with newer research questioning how stable risk preferences are [10, 42].

Trust [43, 44] is defined in economics as the expectation that when in a vulnerable situation, others won't take advantage of it [45, 46]. For example, in a trust game, the trustor gives resources to a trustee, which are tripled by the experimenter (to incentivize the behavior). The trustee can then freely decide whether to give something back to the trustor. The expectation that the trustor will give something back is typically the measure of trust [47]. Trust is particularly important in non-perfect market setups, because contracts in such setups can never be 100% complete and enforcement of contracts is costly. Trust is necessary for actors to be able to rely on their counterparts upholding agreements, and that such agreements will not need to be enforced, thus incurring considerable costs. Trust is desirable, because it can thus reduce transaction costs [48, 49]. Trust (in people) is also strongly correlated with risk preferences [13] but clearly distinct from them [50].

Altruism is defined as the costly act of giving something beneficial to others without expecting anything in return [51, 52]. Altruism is often used to reflect caring about others and their outcomes. Traditionally, economists from the neoclassical economic perspective had been puzzled by the behavior in the so-called dictator game, where completely anonymous individuals decided to share money with an unknown second person they would never know. This behavior has traditionally been called altruism in economics (although later tests refine the understanding as to how much this behavior really does measure a trait of altruism [53, 54]. Altruism has been linked to general pro-social behaviors [55, 56] but also environmentally friendly behaviors [57, 58].

Finally, reciprocity is one of the earliest and arguably most important behavioral economic concepts [59, 60]. It is defined as the willingness to reciprocate an action in kind without enforcement by an institution. Reciprocity is closely related to trust. Whereas trust is the belief or expectation that other people will act in a trustworthy way, reciprocity is more dynamic. Positive reciprocity describes the act of returning a favor. By way of contrast, negative reciprocity has been proposed as a primary driver of sustaining costly punishment of norm violators. Such punishment is described as a major driver of cooperation. For example, both positive and negative reciprocity are important factors shaping workers' behavior towards their employer, either when returning the favor of higher pay with extra effort, or engaging in backlash against lower-than-expected pay [55, 61–63].

Economic preferences are usually not related to each other in studies, that is, each measure is a stand-alone measure often used as an explanatory variable [13, 44]. Therefore, no mapping of the measures similar to that of the values has been proposed so far, to the best of our knowledge.

## Mechanisms underlying possible relations between basic human values and economic preferences

Whereas previous literature has examined the relations between economic preferences and psychological predictors of behavior such as the Big Five personality traits or locus of control [1, 64, 65], to the best of our knowledge there are hardly any studies examining the relations between economic preferences and basic human values (however, see [66]). Some studies in business economics and marketing examined possible connections between values and economic preferences (e.g. [67–69]). However, these studies typically focused on specific subsets of values and/or economic preferences in the marketing context rather than on abstract motivations relevant for a wider range of behaviors. Below we describe potential mechanisms that may account for relations between economic preferences and basic human values, and formulate empirically testable hypotheses about these relations based on the proposed mechanisms.

1. *Risk seeking* (in contrast to risk aversion) represents the willingness of an individual to explore potentially beneficial but risky situations. *Conservation* values emphasize quite the opposite, namely, the importance of safety, preservation of the past, and resistance to change. Thus, *(H1) we expect risk seeking to be negatively associated with conservation.*

*Openness to change* underlines the importance of adventures and excitement in life, autonomy and independence in thoughts and action. Thus, and by way of contrast, *(H2) we expect risk seeking to be positively correlated with openness to change.*

Trust represents the belief in other people's fair behavior. It describes the active process of allowing oneself to be vulnerable to other people's exploitation. Self-transcendence is defined as the importance of being concerned with the needs of other people. People with high levels of self-transcendence prioritize the needs of others, hoping that their helpful actions will not be exploited. Therefore, *(H3) we expect a positive association between trust and self-transcendence.*

By way of contrast, self-enhancement prioritizes individuals' own needs over those of the others. People scoring highly on self-enhancement strive for success and power over others, which may come at the cost of the wealth of others. Considering others as competitors is inherent in this value and may induce distrust. Consequently, *(H4) we expect a negative association between trust and self-enhancement values.*

Both altruism and self-transcendence entail a motivation to support others and give up something for the benefit of others. Thus, *(H5) we expect a positive association between altruism and self-transcendence.* Since the motivation underlying self-enhancement is precisely the opposite, that is, seeking one's own benefit at the cost of others' well-being, *(H6) we expect a negative association between the values of altruism and self-enhancement.*

Reciprocity is a socially relevant motivation. Not only does it represent the importance of doing something good and promoting the well-being of others–it also helps to maintain a social status quo in a functioning society, providing an element of security that favors will not be forgotten, the latter underlying both the values of self-transcendence and conservation. Therefore, *(H7) we expect self-transcendence values* and *(H8) conservation values to be positively associated with positive reciprocity.*

As mentioned earlier, reciprocity is a tool to maintain the social status quo, societal trust, and a functioning society, which are important goals for people who score highly on conservation. However, whereas self-transcendent individuals may be happy to express positive reciprocity, they may be hesitant to express negative reciprocity, as it would go against their motivation to do good. Therefore, *(H9) we expect conservation (but not self-transcendence) to be positively associated with negative reciprocity.*

**Table 1. Summary of hypotheses.**

| Higher order value | Risk seeking | Trust | Altruism | Positive Reciprocity | Negative Reciprocity |
|---|---|---|---|---|---|
| Self-enhancement | | - | - | | + |
| Self-transcendence | | + | + | + | |
| Openness to change | + | | | | |
| Conservation | - | | | + | + |

Negative reciprocity could also be considered an expression of power and control over others. It entails an element of punishing or retaliating against improper or undesired behavior. *(H10) We thus expect self-enhancement values to be positively associated with negative reciprocity*. Table 1 summarizes our hypotheses (S1 Table summarizes them on the lower values level). We preregistered a variation of our expectations (though more detailed on the specific values rather than the higher order value level) and the procedure at the American Economic Association's registry for randomized controlled trials https://www.socialscienceregistry.org/trials/6483) under Scharfbillig [70]. In the next section we will present the methodology underlying our hypotheses, and the empirical examination thereof.

## Methods

### Participants

We collected data using convenience samples in two countries, Poland and Germany: a non-student sample of 212 from Poland (123 females, 89 males, mean age 34.0 years, all of Polish nationality) and a student sample of 127 participants from the University of Cologne, Germany (90 females, 36 males, 1 other, mean age 21.2 years, 120 of German nationality and 8 of a different nationality). See full sample descriptions by country in the Supporting Information, S2, S3 Tables. Convenience samples were sufficient for our purpose, because we were not interested in descriptive evidence but in associations, for which we did not necessarily need representative or particularly diverse samples. Several studies have shown that associations between behaviors or attitudes are similar even when comparing them across convenience samples (including student ones) and representative samples [71]. The sample participants were recruited using the snowball technique, and participation was not remunerated. In order to partially compensate for the limitations of the use of a convenience sample, we examined our hypotheses in the two samples separately to gain external validity for the results. In addition, we controlled for sociodemographic characteristics in our regressions to account for potential compositional effects. Ethical approval was obtained from the Joint Ethics Committee of the University of Frankfurt and the University of Mainz, Germany ("Connecting Basic Human Values and Behavioural Economics"). Written informed consent was obtained.

### Survey design and measurements

To ensure that the approaches were as comparable as possible, the same methodology was used for the data collection in the two samples and for both basic human values and economic preferences. As we are using the method of the global preference survey, several of the economic preferences are–in contrast to economic tradition–asked about, rather than derived from a revealed choice. However, the survey still contains several hypothetical choices and the final preferences are calculated based on them. Values are typically measured using self-rated survey items, so we chose this methodology–also for assessing the economic preferences. The most notable way to measure economic preferences via a survey comes from the experimentally validated preference survey module [8]. Participants in both countries answered an online

survey self-administered in German or Polish, respectively. The survey was administered using the online survey tool, LimeSurvey.

**Basic human values.**   To measure values, we used the German and Polish revised versions of the Portrait Values Questionnaire (PVQ-RR; [19]). The PVQ-RR includes 57 items describing a person in terms of the goals, aspirations, or wishes he or she considers important in life. Respondents answer the questions by comparing how similar the person is to them on a 6-point labeled scale (from 1 "not like me at all" to 6 "very much like me"). Each of the 19 values was measured by three items. The value responses were centered by the individual mean of all values as per Schwartz's instructions (see [72]). Each higher-order value was derived as an average of the responses to its corresponding value questions.

**Economic preferences.**   For the economic preferences we used the validated preference survey module [8], which measures the aforementioned preferences of risk aversion, trust, altruism, and positive and negative reciprocity in a lab-validated measure. For each underlying lab behavior involving real money trade-offs, the authors tested multiple survey questions and selected the most predictive survey item(s) for each measure. For some constructs, multiple items were combined based on a fixed, experimentally tested formula. The economic preferences were then generated following instructions in Falk et al. [8], by combining several questions on an economic preference based on the theoretically derived weights of each question–with the exception of trust, which was based on only one question. The complete survey and the item construction can be found in the supporting material (in English, German and Polish). We used translated versions in German and Polish respectively, as provided on the Global Preferences Survey website (https://www.briq-institute.org/global-preferences/home). The data and syntax used to create the behavioral preferences and value scores are available on Open Science (https://osf.io/v3kqt/?view_only=69213a56704647ae910a10cc70fcada5).

**Sociodemographic characteristics.**   Additionally, we asked all participants to provide information on their age, gender (1 = female, 2 = male, 3 = other), nationality (1 = national of Germany, 2 = national of Poland, 3 = other nationality), and perceived income (on a 4-point scale from "Living comfortably on present income" to "Finding it very difficult on present income"). We produced two dummy variables to represent "national of Germany" and "national of Poland", with "other nationality" as a reference category.

## Results

### Correlations

All the analyses were performed using Stata version 18. Since the findings in the multivariate analyses were similar in the two countries, we are reporting the results for the total sample beginning with the descriptives. We performed a measurement invariance test for the values and find full metric invariance and partial scalar invariance, thus confirming that the values are comparable between the two countries, see S8 Table. We further performed a MANOVA test for sample differences in the main dependent variables, economic preferences and values ($\lambda$ (10,288) = 0.76, p<0.01). We found that whereas there were significant group differences in the human values ($\lambda$ (4,325) = 0.77, p<0.01), economic preferences were rather similar ($\lambda$ (6,293) = 0.99, p = 0.99). However, the overall proportion of the variance attributable to the group difference is only about a quarter. In the pooled analysis we controlled for nationality. S3, S4 Tables provide both the descriptives and the results of the multivariate analysis for the two samples separately. Table 2 presents the means and standard deviations of the higher order values and the economic preferences for a pooled sample of the two countries.

S1 Fig in the Supporting Information shows the correlations between values and economic preferences overall and per country. We see clear correlations profiles which seem to be mostly

**Table 2. Descriptive statistics.**

| Variable | N | Mean | Sd. | Min | Max |
|---|---|---|---|---|---|
| *Sociodemographics* | | | | | |
| Age | 339 | 29.24 | 12.01 | 17.00 | 84.00 |
| Gender | 339 | 1.37 | 0.49 | 1.00 | 3.00 |
| Income | 327 | 1.76 | 0.65 | 1.00 | 4.00 |
| *Higher Order Values* | | | | | |
| Self-Enhancement | 338 | -1.04 | 0.81 | -3.26 | 1.18 |
| Self-Transcendence | 338 | 0.63 | 0.46 | -0.79 | 1.86 |
| Openness to Change | 338 | 0.34 | 0.49 | -1.72 | 1.48 |
| Conservation | 338 | -0.20 | 0.44 | -1.49 | 0.86 |
| *Economic Preferences* | | | | | |
| Risk taking | 336 | 0.00 | 0.79 | -2.73 | 2.41 |
| Positive Reciprocity | 335 | 0.00 | 0.77 | -3.35 | 1.14 |
| Negative Reciprocity | 327 | 0.00 | 0.80 | -1.89 | 3.14 |
| Altruism | 339 | 0.00 | 0.82 | -3.02 | 2.01 |
| Trust | 321 | 0.00 | 1.00 | -1.85 | 1.84 |

Notes: Sample statistics for the pooled sample (Germany and Poland aggregated). Only one participant selected "other" as a response to the gender question and is coded as a 3 here; female is 1 and male is 2. Sd = standard deviation. Values are centered at the individual level as per Schwartz' [73] instructions. Economic preferences are standardized based on the instructions in Falk et al. [8].

consistent in each of the two countries. The Pearson correlations between economic preferences and values were weak or moderate [74]. S5 Table presents the Pearson correlations between the four higher order values and the six economic preferences. In line with H1 and H2, openness to change was significantly positively correlated ($r = 0.23$, $p < 0.000$) and conservation significantly negatively correlated ($r = -0.24$, $p < 0.000$) to risk seeking. In other words, individuals attributing higher importance to the value of openness to change also exhibited a stronger risk seeking preference, while individuals attributing higher importance to conservation values exhibited a weaker risk preference.

In line with H3 and H4, trust was correlated positively to self-transcendence ($r = 0.16$, $p = 0.01$) and negatively to self-enhancement ($r = -0.25$, $p < 0.00$). It was also correlated positively to conservation values ($r = 0.14$, $p = 0.01$) and negatively to openness to change values ($r = -0.11$, $p = 0.06$), for which we had not proposed any hypotheses.

The associations of higher order values with altruism provided empirical support for H5 and H6. Self-enhancement was negatively correlated to altruism ($r = -0.28$, $p < 0.00$), and self-transcendence positively ($r = 0.32$, $p < 0.00$). In other words, individuals attributing higher importance to self-enhancement values were less likely to display altruism, whereas those attributing higher importance to self-transcendence values were more likely to display altruistic preferences. Altruism also displayed a negative correlation to openness to change values ($r = -0.19$, $p < 0.00$) and a positive correlation to conservation values ($r = 0.10$, $p = 0.06$), for which we had not proposed any hypotheses.

The correlation between self-transcendence and positive reciprocity was positive ($r = 0.16$, $p < 0.00$) but its correlation with conservation was insignificant ($r = 0.04$, $p = 0.46$). Thus, empirical support was provided for H7 but not for H8.

Finally, negative reciprocity displayed, as expected in H9, a positive association with self-enhancement values ($r = 0.26$, $p < 0.00$). However, and in contrast to H10, it was negatively correlated with conservation values ($r = -0.18$, $p = 0.00$).

In order to correct for multiple hypothesis testing, the conservative Bonferroni approach would require adjustment of the significance level to 0.005 (0.05/10). With this correction, out of the above 8 supported hypotheses, only H4 had to be rejected–and even then only just ($p$ = 0.0056). S1 Text also shows the analysis for correlations per country separately.

## Multivariate analysis

Next, we delved deeper into the relationship between values and economic preferences. Correlations may be subject to multiple omitted variables bias. One way to address this is to use a regression analysis with all theoretically relevant values as predictors simultaneously, while controlling for sociodemographic characteristics. We avoided including higher-order values in the regressions that were not hypothesized to be associated with the respective economic preference, as this would likely have caused a multicollinearity problem. The reason for this is that values are theorized to span over a two-dimensional circular space [51]. All values that were hypothesized to be related to a specific economic preference were included in the regression. Furthermore, although we used economic preferences as dependent variables and values as predictors, we did not assume any causal relationship between them, but rather merely associations. While there were some differences in effects sizes in the reversed regressions, none of the associations was strong, lending further support to the result suggesting that the values and the economic preferences do not show considerable overlap. Table 3 presents the regression coefficients of these models (S7 Table provides the result for lower level values, S2 Fig show the regression lines in a scatter plot).

Supporting *H2*, risk seeking displayed a negative and significant association with conservation. Openness to change was no longer significant, but according to the motivational trade-offs between openness to change and conservation, it is unlikely that both values would be simultaneously significant in the same regression as they are located on opposite sides of the value circle. The second regression for trust provided support for *H4*. The regression showed a negative and significant association between trust and self-enhancement. Also in this case, the lack of positive and significant correlation with self-transcendence was likely a result of the trade-off between self-enhancement and self-transcendence, which are located in opposition to each other on the value circle. In line with *H5*, altruism turned out to be positively and significantly associated with self-transcendence, but not with the opposing value of self-enhancement.

**Table 3. Standardized OLS regression coefficients.**

|  | Risk Taking | | Trust | | Altruism | | PosReci | | NegReci | |
|---|---|---|---|---|---|---|---|---|---|---|
| **Openness to Change** |  | *0.17 (0.18)* |  |  |  |  |  |  |  |  |
| **Conservation** |  | *-0.41\*\* (0.20)* |  |  |  |  |  | *0.01 (0.13)* |  | *-0.14 (0.18)* |
| **Self-Transcendence** |  |  |  | *-0.00 (0.18)* |  | *0.51\*\*\* (0.15)* |  |  |  | *0.12 (0.19)* |
| **Self-Enhancement** |  |  |  | *-0.24\*\* (0.10)* |  | *-0.04 (0.08)* |  | *-0.11 (0.07)* |  | *0.29\*\*\* (0.11)* |
| **Controls** | *YES* | *YES* | *YES* | *YES* | *YES* | *YES* | *YES* | *YES* | *YES* | *YES* |
| **N** | *324* | *324* | *308* | *308* | *326* | *326* | *322* | *322* | *315* | *315* |
| **R** | *0.02* | *0.10* | *0.06* | *0.09* | *0.10* | *0.17* | *0.06* | *0.07* | *0.03* | *0.10* |

Notes: The dependent variables are in the columns. Heteroscedasticity robust standard errors are in brackets. Significance levels

\* p < 0.1

\*\* p < 0.05

\*\*\* p < 0.01. Sociodemographic controls (not shown) included gender (female as reference category, male and "other"), nationality (Polish as reference category, German and "Other nationality" representing individuals who were nationals of neither Poland nor Germany), age, and income. PosReci = positive reciprocity; NegReci = negative reciprocity; N = sample size. S6 Table shows the regression table with the coefficients for the control variables.

Positive reciprocity did not display any significant association with self-transcendence or conservation when included together in the model. However, negative reciprocity and self-enhancement (but not conservation) showed a positive and significant association, lending support for H10.

## Summary and discussion

Both Schwartz's basic human values approach and the economic preferences approach have been developed and applied to explain human behavior in various domains by measuring people's underlying preferences and motivations. Both approaches have been applied successfully in thousands of studies, albeit in isolation from each other, the first mainly in social psychological studies, and the second mainly in economic studies. We know only little about whether and to what extent concepts from both approaches are in fact mutually exclusive or rather complementary for explaining behavior. Findings suggesting that they are not one and the same may be very relevant for studies attempting to explain behavior. They could imply that combining both approaches to empirically explain behavior in a theory-driven way might be beneficial, because values and economic preferences may tap into different motivations of human behavior.

The present study has attempted to fill this gap by examining the relationships between basic human values and economic preferences. We examined the associations between the higher-order values of self-transcendence, self-enhancement, openness to change and conservation from Schwartz's value theory and the economic preferences of risk aversion/seeking, altruism, trust, and positive and negative reciprocity. We proposed mechanisms as to how they may be associated. For example, we expected a positive association between conservation and risk aversion as both are motivated by attributing importance to stability and the status quo, or between self-transcendence and altruism, as both are motivated by concern for others. For the empirical analysis we employed convenience samples collected in Poland and Germany that assessed both values and economic preferences. The results largely supported our expectations: Several basic human values and economic preferences displayed associations in line with our hypotheses. However, their correlations were only slight or moderate, supporting the idea that values and economic preferences are not mutually exclusive but may rather be complementary. The two approaches reflect motivations that differ to some degree. Indeed, the strongest correlation we found was 0.32 between altruism and self-transcendence. Even for more specific human values, the highest correlation we found was 0.28, between stimulation and risk taking. This suggests that the correlations were not method-specific, and both when considering higher-order values or more specific values, associations exist, but they are not strong. Some but not all of these findings were corroborated in OLS regression analyses while controlling for sociodemographic variables and possible compositional effects. Indeed, values in Schwartz's theory are highly correlated with each other. Therefore, it is not surprising that when including several of them in the same regression, they did not simultaneously display a significant association with economic preferences. Those associations in the regression models which were significant were in line with our expectations. Economic experiments that are used to reveal behavior often incentivize choices of participants with money, and this is also kept as a hypothetical incentive in the survey method for economic preferences. This means, though, that there is likely a specific dimension remaining underexplored in this area. Money–or, more precisely, having more money–represents a specific value (power wealth) within the self-enhancement higher-order value. Thus, everything incentivized with money will automatically address the value of self-enhancement and oppose the self-transcendence values like universalism (e.g. care, nature). This tradeoff is well established in economics when it comes to conflicts

around allocations and other-regarding preferences [53, 75]. However, tradeoffs between conservation values and openness values are less represented in the economics research. While this makes sense for many of the topics of interest in economics, like labor markets, macroeconomic development, competition, firm behavior and related regulations, fundamental issues that are important to people also politically, especially in the realm of migration and immigration, innovation and its adoption, policing and justice systems, and international relations will be likely overlooked [76]. In other words, conflicts between self and other are different from conflicts between self and openness (new ideas, exploration, diverse choice) and self and conservation (ingroup focus, security, traditions). Similarly, measuring risk preferences in the monetary dimension as an absolute measure is not a good proxy for risky behavior in areas that relate to interaction with other people, which is in line with findings in other studies [77].

The findings also follow on from a recent number of papers that criticize measures in the economic literature that are gathered by way of economic games, because they seem to suffer from low external validity [42, 78]. Wang & Navarro-Martinez [79] show that one-shot behavioral measures typically used in economic research are weak predictors for behaviors in the field gathered through surveys and suggest using multiple measures instead. That means, instead of using the outcome of one specific game, for example the strategy chosen in a prisoner's dilemma, they advised using responses from several experiments (either the same one repeated or different theoretically related ones). While the survey methods used to elicit economic preferences in this paper are already combinations of several questions, they are still calibrated on experimental laboratory games Falk, Armin, et al. "The preference survey module: A validated instrument for measuring risk, time, and social preferences." (2016). We show that our economic measures are already connected to several values measures and as such, the question is whether an aggregation of economic measures captures values content rather than better preferences, or vice-versa.

It is also debatable whether the neutrality condition of monetary rewards for generating experimental evidence overall is always fulfilled. The typical but mostly unstated assumption in economic experimental research is that using monetary rewards is the best solution, because even if participants do not want to receive money, they can always use the money to buy whatever they desire. Thus, the reward would be beneficial to all and never achieve a level of satiation where participants would no longer care about getting more out of the experiment. This contrasts, for example, with other ways to incentivize the experiment such as using chocolate or alcoholic beverages, which not everyone would like and where at some point the reward would no longer be interesting. Knowing that monetary incentives fall in one specific area of the values circle, one could argue that one of the fundamental scientific criteria for validity is violated. While we would not go that far in our assessment, it at least leaves open the question whether certain trade-offs that have been studied so far would result in different outcomes if the method for incentivisation did not fall into the value realm of self-enhancement (e.g. money, status, reputation) but rather into areas of conservation (strengthening traditions, enforcing norms, satisfying security needs, health) or openness to experience (ability to explore new things, being rewarded with something unique, doing something creative, satisfying curiosity, etc.).

The results are in line with the findings of Becker et al. [1], who found low to moderate correlations between personality and economic preferences, both acting more as complements than substitutes in predicting behavior. Given the relevance of values for attitudes and behavior, it thus seems at least plausible to consider using both in the study design.

There are several limitations to our study. First, our findings rely on convenience samples. Indeed, participants in our study may not be representative of their respective populations. However, our goal was to not to describe populations but rather to carry out an empirical

detection of theoretically driven associations. To partially address the samples' limitation, we used samples from two different countries. The results in each sample were essentially similar, lending further confidence in the findings. Second, we measured economic preferences using a survey rather than relying on revealed preferences of participants in their behavior. While our approach may lead to different measures of individuals' economic preferences, it has the advantage that we used the same methodology to collect data on basic human values and economic preferences, thus limiting the risk of method effects creating bias within our measures and the associations thereof [80]. Finally, our study focused on the measurement and estimation of associations between values and economic preferences rather than in examining how they may operate in tandem for explaining behavior in various domains. Indeed, future studies may include both behaviors, attitudes toward different social groups or policies, and both basic human values and economic preferences in an attempt to provide more conclusive theory-driven behavioral or attitudinal explanations.

Notwithstanding these limitations, our study is–to the best of our knowledge–the first to conduct an empirical assessment of theory-driven associations between abstract basic human values and economic preferences. Our findings suggest that while they share some variability, they are not mutually exclusive. As such, they may be rather complementary, and they may have their unique contribution when used in concert to explain attitudes or behavior.

## Supporting information

**S1 Fig. Correlations between values and economic preferences.**
(PNG)

**S2 Fig. Scatter plot of values and economic preferences.** Scatter plot on values and economic preference. Grey squares represent observations, red line linear fit regression lines, grey shaded area 95% confidence interval.
(PNG)

**S1 Table. Preregistered hypotheses on first-order values.**
(PDF)

**S2 Table. Descriptive statistics of the specific lower-level values.**
(PDF)

**S3 Table. Descriptive statistics–Polish sample.**
(PDF)

**S4 Table. Descriptive statistics–German sample.**
(PDF)

**S5 Table. Variable correlations.** Pearson correlation coefficients (top) and significance levels (bottom).
(PDF)

**S6 Table. OLS regression coefficients with all socio-demographics as control variables.** The dependent variables are in the columns. Heteroscedasticity robust standard errors are in brackets. Significance levels: * $p < 0.1$, ** $p < 0.05$, *** $p < 0.01$. Females were the reference category for gender, the variable "Other Nationality" represents individuals who were not country nationals, with Polish nationals as the reference category. PosReci = positive reciprocity; NegReci = negative reciprocity; N = sample size.
(PDF)

**S7 Table. Predicting behavioral preferences—lower level values.** OLS regression results using top row variables as dependent variables. Heteroscedasticity robust standard errors are in brackets. Significance levels: * p < 0.1, ** p < 0.05, *** p < 0.01. Variable names: UNC–Universalism concern, UNN–Universalism nature, UNT–Universalism tolerance, POD–Power dominance, POR–Power resources, BEC–Benevolence care, BED–Benevolence dependability, AC–Achievement, SEP—Security personal, SES–Security societal, COR–Conformity rules, COI–Conformity interpersonal, TR–Tradition, ST–Stimulation, SDA–Self-direction action, SDT–Self-direction thought.
(PDF)

**S8 Table. Measurement invariance of higher-order values across Germany and Poland.**
(PDF)

**S1 Text. Country differences.**
(PDF)

## Author Contributions

**Conceptualization:** Mario Scharfbillig, Jan Cieciuch, Eldad Davidov.

**Data curation:** Mario Scharfbillig, Jan Cieciuch.

**Formal analysis:** Mario Scharfbillig, Jan Cieciuch.

**Methodology:** Mario Scharfbillig, Jan Cieciuch.

**Software:** Jan Cieciuch.

**Writing – original draft:** Mario Scharfbillig, Eldad Davidov.

**Writing – review & editing:** Mario Scharfbillig, Jan Cieciuch, Eldad Davidov.

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
