## [Decision Letter · Decision Letter 0]

14 Apr 2023

PONE-D-23-03410One and the same? How similar are basic human values and behavioral economic preferencesPLOS ONE

Dear Dr. Scharfbillig,

Thank you for submitting your manuscript to PLOS ONE. After careful consideration, we feel that it has merit but does not fully meet PLOS ONE’s publication criteria as it currently stands. Therefore, we invite you to submit a revised version of the manuscript that addresses the points raised during the review process. As you can see from the reviewers’ comments below, they do see the value of your project but it will take several major revisions for the paper to be accepted for publication. 

First of all, two reviewers had issues with the motivation of the paper. Please strengthen this part by providing some rationale for why it is important to compare the values approach (top-down) with the behavioral economics approach (bottom-up) . Especially, at the conceptual level, one the reviewers is especially critical of how the paper uses the term “behaviroal“ in a very unconventional sense. Please also offer some justifications or make revisions accordingly.

Second, all reviewers raised concerns about the sample characteristics of your studies. For example, since they are not incentivized economic experiments, why did you simply recruit students? In addition, since the data was collected from Germany and Poland, is there a systematic difference between these two country samples? Can they be simply pooled for analysis?

Third, analytically, all reviewers raised several questions about both the statistical strategies and the paper’s explosion of the results. Please provide more detailed justifications for your methods and improve how you report the results to avoid potential confusions.

Finally. there are also quite a few minor suggestions made by all the reviewers. Please also address them properly.

We look forward to receiving your revised manuscript.

Kind regards,

Hans H. Tung

Academic Editor

PLOS ONE

Journal Requirements:

3. We note that Figure 1 in your submission contain copyrighted images. All PLOS content is published under the Creative Commons Attribution License (CC BY 4.0), which means that the manuscript, images, and Supporting Information files will be freely available online, and any third party is permitted to access, download, copy, distribute, and use these materials in any way, even commercially, with proper attribution. For more information, see our copyright guidelines: http://journals.plos.org/plosone/s/licenses-and-copyright.

(1) You may seek permission from the original copyright holder of Figure 1 to publish the content specifically under the CC BY 4.0 license. 

6. Please ensure that you include a title page within your main document. You should list all authors and all affiliations as per our author instructions and clearly indicate the corresponding author.

7. Please include a separate caption for each figure in your manuscript.

Reviewers' comments:

Reviewer's Responses to Questions

**Comments to the Author**

1. Is the manuscript technically sound, and do the data support the conclusions?

Reviewer #1: Yes

Reviewer #2: Partly

Reviewer #3: Yes

2. Has the statistical analysis been performed appropriately and rigorously? 

Reviewer #1: Yes

Reviewer #2: Yes

Reviewer #3: Yes

3. Have the authors made all data underlying the findings in their manuscript fully available?

Reviewer #1: No

Reviewer #2: Yes

Reviewer #3: No

4. Is the manuscript presented in an intelligible fashion and written in standard English?

Reviewer #1: Yes

Reviewer #2: Yes

Reviewer #3: Yes

5. Review Comments to the Author

Reviewer #1: I paste my report below and upload it as an attachment as well.

Review report for PONE-D-23-03410, “One and the same? How similar are basic human values and behavioral economic preferences”

This manuscript attempts to address an interesting research question. Because psychologists measure basic human values and economists measure preferences, the authors would like to see whether values and preferences are similar and study the possible mechanism underlying the similarity.

To do that, they use surveys to measure mainly four values (openness to change, conservation, self-transcendence, and self-enhancement) and five preferences (trust, risk preferences, positive reciprocity, negative reciprocity, and altruism) of 339 participants in Poland and Germany and proceed in three steps.

In the first step, the authors calculate the correlation coefficients between each individual value and each preference. Not surprisingly some correlations are significant (for instance, openness to change is positively correlated with risk taking) but others are not. In the second step, the authors regress preferences on values but at the same time, they control for some sociodemographic characteristics. Again, not surprisingly, some values can explain preferences even when the sociodemographic characteristics are controlled for, but others cannot. Lastly, they offer a visualization regarding how each particular preference is close to a particular value. Overall, the research question is interesting, but the results are not surprising. Presumably, even if we just focus on preferences, we may very likely arrive at the same conclusion as some preferences may be more similar to each other (for instance, trust and altruism are quite similar) and others are less so. Hence, at the end of the day, it is not clear what additional insight we have learned from this research. The mechanism, which interests me the most when reading the abstract, is lacking.

With that said, because Plos One only asks for scientifically valid research, as far as this is concerned, with some revisions, the manuscript would meet the publication requirement.

I now bullet list comments/questions/suggestions I have. The manuscript does not number pages, but page number should be easily inferred.

(1) Throughout the manuscript, the authors more often than not use the term behavioral economics in a rather unusual way. For instance, risk preferences are quite classical preferences, it is unusual to describe them as behavioral. The term “behavioral” often refers to preferences that are not classical and hence insights from psychology or cognitive neuroscience are used to motivate why economists should care about them.

(2) On page 3, second paragraph, line 5, the sentence “While the values approach is characterized by a top-down approach, thinking about the highest possible motivation level people can have that then influences behavior, the behavioral economic approach operates in the opposite way” sounds strange. Please check your grammar. The next sentence, “It starts from the behavior and tries to infer the preference structure using a revealed choice approach” has a rather unusual way of describing what economists do. Economists infer preference from choice, this inference is called the revealed preference approach (not revealed choice approach). Choice is the data of economists; it is preference that gets revealed from choice. For economists, preference is the underlying driver of choice much like values drive behavior for psychologists. Since the authors use the Global Preference Survey (GPS) developed by Falk et al. and there are survey questions that directly ask about preferences, it is not clear there is indeed such a difference between these two approaches.

(3) On page 6, why is the section titled “Behavioral economic preferences” shorthanded as BPP? Again, the use of the term “behavioral” is unusual. Even Falk did not call the survey module behavioral.

(4) On page 6, 9th line from the bottom, “the risk premium” probably should be “the risk premium of stocks.”

(5) On page 8, 7th line from the bottom, the word “I” should be removed.

(6) On page 12, now BPP refers to “behavioral economic preferences parameters.” In this paragraph, the authors claim the complete survey and item construction can be found in the Supporting Information (in English). It is not included there. Also, the authors claim they will make data and the syntax available on osf.

(7) On page 13, the authors pooled data from Poland and Germany together. A formal statistical test of no difference is needed before data from the two countries can be pooled.

(8) The authors should keep the reported values to the same decimal places. Sometimes they report up to 2, other times to 3.

(9) The authors should make sure the values they reported in the table are the same as what they reported in the text. Just to give one example, on page 15, 2nd line, they reported an r=-0.283 but the corresponding number in Table 3 is -0.284. There are too many inconsistencies. The burden should fall on the authors, not on the readers.

(10) When the authors said there is a weak correlation, do they base the claim on a small r or a small p-value? The authors should explain their criterion and stick to it consistently throughout.

(11) The authors did not report the results for H7 and H8. But in fact, positive reciprocity does correlate with self-transcendence. Why do they ignore this result? Positive reciprocity also negatively correlates with self-enhancement, by the way.

(12) The author should justify why they do not do multiple comparison corrections for the results they report in Table 3. If the corrections are made, some significant results may become insignificant.

(13) Shaded color should be explained in the legends of Table 3.

(14) GPS does include preferences on patience. The authors do not mention patience throughout the manuscript but somehow there is a column on patience in Table 4.

(15) On page 20, the last paragraph the authors seem to expect that results in step 2 would be similar to those in step 1. I do not understand their logic. They only calculate correlation coefficients in step 1. But in step 2, they run multiple variable regressions. Logically there is no guarantee that one implies the other. So why should one expect to see similar results in both steps? Because the results are not robust, the authors should address the conceptual difference among the three steps to help the readers understand which step best addresses their research curiosity.

(16) Table A5 is confusing. I don’t understand what “Same Sample all” means though I did notice that the sample size is fixed at 296.

(17) The title of Table A6 should be changed. The authors are regressing values on preferences here.

(18) The shorthands for lower-level values in Table A8 should be explained.

(19) On page 39, several p-values and table number are missing (see XXX and x).

(20) Currently all figures are called Fig. 1. See pages 43, 44.

(21) Again, the shorthands for lower-level values on page 44 should be explained.

Overall, the manuscript is not written with enough care. The authors should read the manuscript before they submit their revised version.

Reviewer #2: PLOS ONE review – PONE-D-23-03410

“One and the same? How similar are basic human values and behavioral economic preferences?” M. Scharfbillig, J. Cieciuch and E. Davidov

This manuscript addresses the link between basic human values and behavioral economic preferences. The authors are the first to investigate this link. Behavioral economics is an interdisciplinary field of study that incorporates insights from psychology and other social sciences to understand how individuals make decisions. In contrast to traditional economics, behavioral economics does not assume that people make rational decisions. People can be irrational and do not always go for the maximum utility. In contrast, research on basic human values in social psychology focuses on underlying psychological motivations of people. The broad field of behavioral economics and the specialized field of human values have not been connected before. In economics, basic human values have been used in business economics such as in marketing (see work by Steenkamp and colleagues); in general economics human values are, as far as I know, hardly used.

The topic is interesting, but its execution has several weaknesses. These concern both the theoretical framework and the methodology. Several issues, I think, are not yet well developed in the current manuscript. In addition, the layout is at times a bit sloppy.

The focus in the article is on the study of human values and that part of the manuscript is well developed. However, in the literature on behavioral economics mainly older references are used and I miss more recent additions to this literature such as the work by e.g., Thaler and Camerer.

Page numbers are missing in the manuscript, and I numbered them myself for convenience. I started counting at the page entitled “Introduction”. I refer to my own page numbers in the sequel.

On p. 2 it is stated that the values approach works top-down and the behavioral economics approach bottom-up. This may be (is) true, but why is this a reason to bring these streams together? Who will benefit? The economics literature or the values literature? Please elaborate.

On p. 3 it is stated that the fields use “fundamentally different methods”. Is that true? In more applied fields such as marketing a behavioral economics approach is taken employing experiments and surveys which are also part of the values’ researcher’s toolbox. Please clarify what the differences are. Later in this page it is stated that the respective literatures will be enriched, but how? What may it bring to economists or values researchers?

On p. 7 the hypotheses on e.g., risk, trust, and altruism are not novel for researchers in business economics (see e.g., Keh & Sun, 2008 in JIM; Kumar et al., 2021 in JRCS; Ahmad et al., 2023 IMR).

On p. 10 in the method section the samples are different in age which may result in the samples that differ in the importance of and the variance in their values. Are these students from the same faculty? It is known that topic of study affects values (Bardi et al., 2014). Please provide more information on the students’ background.

P. 10, survey design. The samples are from different countries. Please provide information on assessment of common method variance and invariance testing when appropriate. I think this holds at least for part of the Falk et al. survey.

p. 11. Why is the focus on the four higher-order values?

It worries me that in Table 4 the effect of “Other” is large and significant. What does it stand for?

On p. 17 the focus becomes on MDS. It is not clear what type of MDS is used. Please elaborate on this. The authors mention that they use MDS to visualize the results. Visualization is a nice feature of MDS, but why is the visualization done? What insight does it provide that helps us understand the importance of linking the behavioral economics approach to the values approach?

In the summary and discussion, I again get confused on who is the target audience of this manuscript. Is it behavioral economists or values researchers, or both? I miss a clear statement on what can be learned from the current research. You may bring results from Table A7, A9 and A10 to the main text, I think this appeals more to economists. Finally, please elaborate the discussion including recent work from both values research and behavioral economics.

Minor issues

p. 4 last sentence. Value do not “determine” attitudes. There is a correlation, however, there is no causality.

p. 12/13 Please provide information on Germany and Poland separately in the descriptive statistics. Also clarify whether the centring was done per country. The “0” on all dependent variables suggests otherwise.

p. 12 Please change “Gender” in “male or “female” for clarity. Please also check the effect on outcomes when the “3” for gender is excluded. I do not expect many differences, but an outlier may affect results.

p. 13 Please provide more insight into the data. How is the distribution of e.g., risk?

p. 13 why is it unexpected that trust is positively related to conservation? The context of the questions may affect this result (e.g., being in a well-known traditional environment may breed trust)

p. 14 relationship altruism conservation and openness. Why unexpected? In the charitable giving literature this result is common.

p. 18 Figures 2 and 3 are not in the paper. All Figures are titled “Figure 1”. Likely a typing error.

p. 20 last paragraph. Please add references to substantiate the points made.

Reviewer #3: Summary: To explain the drivers of human behavior, the economics literature has emphasized concepts such as risk-taking, trust, altruism, and positive/negative reciprocity. On the contrary, the social psychology literature has emphasized the role of values (i.e., abstract principles and goals in life) such as self-transcendence vs conservation and self-enhancement vs. openness. However, little is known about how these two are empirically related to each other because these two distinctive literatures have been developed separately. To fill this gap, the authors conduct a survey wherein subjects' basic human values and behavioral economic preferences were measured simultaneously. The authors found a statistically-significant correlation between these two concepts, but the correlation is moderate, suggesting that both capture a significant unique content.

Evaluation: It is often the case that psychologists and economists measure different yet somewhat similar concepts in different ways. How they are related (whether they eventually measure the same or not) is a natural and important question. I acknowledge that the authors tackle this question. Behavioral economic preferences are properly measured in the standard way a la Falk et al. (2018, QJE) (I'm an economist; thus, I leave judgment on the way to measure human basic values). The empirical analysis is simple and transparent; it is clearly stated that they analyze just correlation (not causal effect). Based on these considerations, I think that this paper will contribute to the literature on economics and social psychology; thus, I recommend R&R.

Comments:

1. As behavioral economic preferences, the authors measure risk-taking, altruism, trust, and positive/negative reciprocity. I agree that they are important factors, but other factors also have been identified as key factors in economic decision-making. A prominent example is time preference (i.e., how patient an agent is). It would be better to explain the reason why the authors focus on risk-taking, altruism, trust, and positive/negative reciprocity (perhaps, is it because time preference seems to be unrelated to values?)

2. Related to the first point, the rightest column in Table 4 contains patience as a dependent variable. Does it mean that the survey module includes questions on time preference? If so, this point should be explained and the analysis of them should be included in the main text. In addition, would there be some other behavioral economic preferences measured but not explicitly mentioned in the paper? If so, they should be also explained.

3. The authors mention "the complete survey and the item construction can be found in the Supporting Information," but I could find them in the supporting information. The availability of the survey questions is important to enhance the replicability of the research. It would be better to present them in the supporting information. (In case it was presented but I missed them, I'm very sorry for it).

4. I missed how incomes were coded. According to Table 2, it seems that they were measured on a 6-point scale, but it would be better to explain the details (I found how gender was coded).

5. A sample in Germany is the student sample, while that in Poland is not the student one. Is the latter one a representative sample? How were they recruited? It would be better to explain these points. Related to these points, I have several questions (it would be better to explain them either in the main text or in the supporting info.):

a.) I understand that this survey is not a monetary incentivized one, but did you pay some money for completing the survey or was it perfectly volunteered?

b.) At the beginning of the method section, the authors mention "a student sample from the University of Cologne". I expect that this is a German sample, and it might be better to explain this explicitly.

c.) In the conclusion, the authors mention "There are several limitations to our study, the first of which being that we are making inferences

using student samples." From this sentence, one might have an expression that both Poland and German samples were students. It might be better to note that the Polish samples were not students.

d.) If this survey were a monetary-incentivized laboratory experiment, I would understand that only student samples are convenient. However, this is just a survey and convenient samples that are more diverse socio-demographically (e.g., amazon M-Turk samples in the US). It might be useful to explain why the authors chose student samples.

6. In addition to presenting correlation coefficients, it might be also useful to show scatter plots where the x-axis is BPP and the y-axis is value, because it visualizes that the correlation is only moderate.

I enjoyed reading the manuscript. Thank you so much.

6. PLOS authors have the option to publish the peer review history of their article (what does this mean?). If published, this will include your full peer review and any attached files.

Reviewer #1: No

Reviewer #2: No

Reviewer #3: No

---

## [Author Response · Author response to Decision Letter 0]

17 Nov 2023

Please note the separate file we have uploaded detailing all answers to specific questions.

---

## [Decision Letter · Decision Letter 1]

7 Dec 2023

PONE-D-23-03410R1One and the same? How similar are basic human values and economic preferencesPLOS ONE

Dear Dr. Scharfbillig,

Thank you for submitting your manuscript to PLOS ONE. The reviewer with substantive concerns about your original manuscript has agreed that most of the issues have been properly addressed and the reviewer has some minor suggestions. Please revise your manuscript accordingly. 

We look forward to receiving your revised manuscript.

Kind regards,

Hans H. Tung

Academic Editor

PLOS ONE

Journal Requirements:

Reviewers' comments:

Reviewer's Responses to Questions

**Comments to the Author**

1. If the authors have adequately addressed your comments raised in a previous round of review and you feel that this manuscript is now acceptable for publication, you may indicate that here to bypass the “Comments to the Author” section, enter your conflict of interest statement in the “Confidential to Editor” section, and submit your "Accept" recommendation.

Reviewer #3: (No Response)

2. Is the manuscript technically sound, and do the data support the conclusions?

Reviewer #3: (No Response)

3. Has the statistical analysis been performed appropriately and rigorously? 

Reviewer #3: (No Response)

4. Have the authors made all data underlying the findings in their manuscript fully available?

Reviewer #3: (No Response)

5. Is the manuscript presented in an intelligible fashion and written in standard English?

Reviewer #3: (No Response)

6. Review Comments to the Author

Reviewer #3: I appreciate the authors' effort in revising the manuscript. I think the points raised by my referee report were addressed adequately. 

There are several minor comments:

1. Abstract "they are equivalent to each other and mutually exclusive, or are they rather complementary": In mathematics, A and B are mutually exclusive if  A ∩ B is empty. Therefore, saying that economic preferences and values are mutually exclusive seems to indicate that they are totally different concepts. However, in the above sentence, it is used to indicate that they are similar concepts. This is quite confusing for me. 

2. Page 5, second paragraph: "We try to)" -> "We try to"

3. Page 8, footnote 2: "preference patience" -> "preference for patience"  "we did not" ->". Thus, we did not"

4. Page 13 "We collected data using convenience samples in two countries, Poland and Germany: 212 from Poland (123 females, 89 males, mean age 34.0 years, all of Polish nationality) and a student sample of..." It would be better to explicitly mention that the Poland sample is not a student sample.  

5. Page 13: In the reply letter, the authors said "We were not interested in descriptive evidence but in associations, for which we did not necessarily need representative orparticularly diverse samples." I agree that this could serve as a justification for using convenience samples in the study. However, this point is not explained in the method section (page 13). It would be better to explain this point. Relatedly, there are several studies reporting that associations between behaviors/attitudes are similar even when we compare convenience samples (including student ones) and representative samples (e.g., Snowberg, E., & Yariv, L. (2021). Testing the waters: Behavior across participant pools. American Economic Review, 111(2), 687-719.) Citing such existing studies may be worthwhile.

I really enjoyed reading the manuscript.

7. PLOS authors have the option to publish the peer review history of their article (what does this mean?). If published, this will include your full peer review and any attached files.

Reviewer #3: No

---

## [Author Response · Author response to Decision Letter 1]

19 Dec 2023

Please see the attached document with the specific responses to the comments. We addressed all of them.

---

## [Editor Report · Decision Letter 2]

20 Dec 2023

One and the same? How similar are basic human values and economic preferences

PONE-D-23-03410R2

Dear Dr. Scharfbillig,

We’re pleased to inform you that your manuscript has been judged scientifically suitable for publication and will be formally accepted for publication once it meets all outstanding technical requirements.

Kind regards,

Hans H. Tung

Academic Editor

PLOS ONE